

# Interference with the retinoic acid signalling pathway inhibits the initiation of teeth and caudal primary scales in the small-spotted catshark *Scyliorhinus canicula*

Isabelle Germon[1], Coralie Delachanal[1], Florence Mougel[1], Camille Martinand-Mari[2], Mélanie Debiais-Thibaud[2] and Véronique Borday-Birraux[1,3]

[1] Laboratoire Évolution, Génomes, Comportement, Écologie, CNRS, IRD, Université Paris-Saclay, Gif-sur-Yvette, France
[2] ISEM, CNRS, IRD, EPHE, Univ. Montpellier, Montpellier, France
[3] Université Paris Cité, Paris, France

Corresponding authors
Mélanie Debiais-Thibaud,
melanie.
debiais-thibaud@umontpellier.fr
Véronique Borday-Birraux,
veronique.borday-
birraux@universite-paris-saclay.fr

## ABSTRACT

The retinoic acid (RA) pathway was shown to be important for tooth development in mammals, and suspected to play a key role in tooth evolution in teleosts. The general modalities of development of tooth and "tooth-like" structures (collectively named odontodes) seem to be conserved among all jawed vertebrates, both with regard to histogenesis and genetic regulation. We investigated the putative function of RA signalling in tooth and scale initiation in a cartilaginous fish, the small-spotted catshark *Scyliorhinus canicula*. To address this issue, we identified the expression pattern of genes from the RA pathway during both tooth and scale development and performed functional experiments by exposing small-spotted catshark embryos to exogenous RA or an inhibitor of RA synthesis. Our results showed that inhibiting RA synthesis affects tooth but not caudal primary scale development while exposure to exogenous RA inhibited both. We also showed that the reduced number of teeth observed with RA exposure is probably due to a specific inhibition of tooth bud initiation while the observed effects of the RA synthesis inhibitor is related to a general delay in embryonic development that interacts with tooth development. This study provides data complementary to previous studies of bony vertebrates and support an involvement of the RA signalling pathway toolkit in odontode initiation in all jawed vertebrates. However, the modalities of RA signalling may vary depending on the target location along the body, and depending on the species lineage.

## INTRODUCTION

Retinoic acid (RA) is an oxidized form of vitamin A and is a pleiotropic signalling molecule known to have many functions during the embryonic development of vertebrates (for a

review see *Ghyselinck & Duester, 2019*). The all-trans form of RA synthesized from carotenoids is a ligand for RA nuclear receptors and they together control the expression of target genes by acting as transcriptional regulators. RA levels are finely controlled during embryonic development, through the balance between RA synthesis by retinaldehyde dehydrogenases (Raldh1 to 3, encoded by the *Aldh1a1* to *3* genes in mammals) and RA degradation by cytochromes P450 26 (Cyp26A1, B1 and C1, encoded by *Cyp26a1-c1* genes in mammals) (for a review see *White et al., 1996*; *Hernandez et al., 2007*; *Roberts, 2020*). Three retinoic acid nuclear receptors (NR1B family: Rar alpha, beta and gamma encoded by *Rara*, *Rarb* and *Rarg* respectively in mammals) interact with retinoid X nuclear receptors (NR2B family: Rxr alpha, beta, gamma encoded by *Rxra*, *Rxrb* and *Rxrg* respectively) (for a review see *Niederreither & Dollé, 2008*; *Fonseca et al., 2020*). The early embryonic RA-dependent signalling pathways described in vertebrates are conserved in chordates (*Fujiwara & Kawamura, 2003*) and conservation of the whole genetic circuitry has also been described at the level of bilaterian animals (reviewed in *Albalat, 2009*; *André et al., 2019*).

The RA signalling pathway has been a versatile, quickly evolving system that participated in the evolution of many derived features of animals. In particular, the RA pathway was shown to be important for tooth development in mammals (*Mark et al., 1995*; *Kronmiller, Nguyen & Berndt, 1995*; *Kronmiller et al., 1995*), and suspected to play a key role in tooth evolution in teleosts (*Gibert et al., 2015*). Teeth are biological structures that present several examples of adaptive morpho-functional changes making them a relevant model in Evo-devo (*Huysseune, Sire & Witten, 2009*). Teeth are present in most jawed vertebrates and show a very high diversity in terms of location, shape or number. Different types of "tooth-like" structures are located at different places of the body, such as the dermal denticles which cover the whole body of chondrichthyans, or the pharyngeal denticles of teleosts (*Sire & Huysseune, 2003*). Despite positional discrepancies, the development of tooth and "tooth-like" structures (collectively named odontodes) seem to be mostly conserved among all jawed vertebrates, both with regard to histogenesis and genetic regulation (reviewed in *Berio & Debiais-Thibaud, 2021*). These conserved features inform us about the ancestral characteristics of tooth development in early vertebrates.

Mouse mutants in RA receptors develop deformed skulls without any teeth (*Mark et al., 1995*). In contrast, embryonic jaws of mice exposed to RA develop additional dental buds and modified tooth identities with molars replaced by incisors (*Kronmiller, Nguyen & Berndt, 1995*). Exposure to exogenous RA or to an increase of endogenous RA by mutation of *cyp26b1* in zebrafish also induce the formation of supernumerary teeth and changes in tooth morphology (*Seritrakul et al., 2012*; *Gibert et al., 2015*). The chemical blockage of the endogenous RA synthesis results in a complete absence of teeth in both mouse and zebrafish, showing that tooth initiation depends on an endogenous source of RA (*Kronmiller et al., 1995*; *Gibert et al., 2010*). In zebrafish the RA signalling pathway appears to be necessary to induce the initial thickening of the pharyngeal odontogenic epithelium, and disruption of this pathway eliminates tooth formation (*Gibert et al., 2010*). However, the formation of teeth in other teleosts such as the medaka and the Mexican tetra is independent of the RA signalling pathway (*Gibert et al., 2010*), questioning the
evolutionary origin of RA signalling function in tooth development (*Jackman & Gibert, 2020*). Although RA signalling in development is ancestral to all vertebrates, and probably originated with metazoans (*Campo-Paysaa et al., 2008*), its implication in tooth development in mammals and some teleost fishes may be convergent and not ancestral to all tooth-bearing vertebrates. Here we explore this issue by testing the putative function of RA signalling in the development of teeth and caudal primary scales in a cartilaginous fish, the small-spotted catshark *Scyliorhinus canicula*. This non-model organism is chosen because: the development of its teeth and caudal primary scales occurs at embryonic stages; embryo collection is easy in rearing facilities; a wide range of sequence data gives access to gene expression patterns (*Reif, 1980*; *Mellinger & Wrisez, 1993*; *Ballard, Mellinger & Lechenault, 1993*; *Debiais-Thibaud et al., 2011*). In this species, the effect of RA signalling interference can thus be tested on both tooth and scale development, and allows intra-organism comparison of the structures without the artefacts generated comparing long-divergent species (*e.g.*, zebrafish *vs* medaka). Finally, comparing RA signalling function in cartilaginous and bony fishes provides a better description of the ancestral status of this signalling pathway in all jawed vertebrates.

## MATERIAL AND METHODS

### Embryo collection

Small-spotted catshark embryos were provided by the Station de biologie de Roscoff, France. All embryos were maintained at 17 °C in sea water at the CNRS animal husbandry facility in Gif-sur-Yvette (facility reference C 91 272 105) until they reached specific developmental stages: few embryos were sacrificed just before treatment to determine total body length. We focused on the development of the caudal primary scales, located at the tip of the tail of stage 29 embryos (about 25 mm body length), and of teeth of the lower jaw of stage late 31/early 32 embryos (about 40 mm body length) (*Mellinger & Wrisez, 1993*; *Ferreiro-Galve et al., 2010*; *Debiais-Thibaud et al., 2011*; *Debiais-Thibaud et al., 2015*; *Cooper et al., 2017*). Embryos were euthanized with MS222 right after the pharmacological treatments or at chosen later stages. The yolk was removed and embryos were fixed 48 h in a phosphate buffer solution containing 4% of paraformaldehyde (PFA) at 4 °C. Embryos were then gradually dehydrated in methanol or ethanol and stored at −20 °C for further analyses.

Handling of small-spotted catshark embryos followed all institutional, national, and international guidelines (European Communities Council Directive of September 22, 2010 (2010/63/UE)) and no further approval by an ethics committee was necessary as the biological material is embryonic.

### Identification of sequences used to generate RNA probes

Four genes of the retinoic acid pathway were first identified in a cDNA library from *Scyliorhinus canicula* (*Oulion et al., 2010*) through a BLAST search with mouse sequences: *aldh1a2* (XM_038813431.1), *cyp26a1* (XM_038774017.1) and *cyp26b1* (XM_038793129.1) were isolated and properly predicted in the recent small-spotted catshark genome available at NCBI (sScyCan1.1 GCF_902713615.1), while an additional *cyp26c1* sequence identified

from the cDNA library could not be identified in the genomic data. However, the best hit for this sequence was the predicted *cyp26c1* gene from another shark, *Carcharodon carcharias* (XM_041209307.1), suggesting missing data from the small-spotted catshark genome.

Six additional coding sequences (*aldh1a1, aldh1a3, rara, rarb, rxrb* and *rxrg*) were identified from a locally assembled small-spotted catshark transcriptome built from small-spotted catshark jaw RNAseq data (Enault et al., 2018) by an identical BLAST search. The sequences for *aldh1a1* (XM_038804963.1), *aldh1a3* (XM_038813505.1), *rara* (XM_038778940.1), *rarb* (XM_038797334.1), *rxra* (XM_038781634.1), *rxrb* (XM_038815745.1), *rxrg* (XM_038793520.1) were predicted transcripts in the sScyCan1.1 genome, although our nucleotide sequences were often shorter than those recorded in NCBI (when different, see Data S1). However, we found no available data for *rarg* in the sScyCan1.1 genome but the best hit with this cDNA extracted sequence was the *rarg* gene annotated in the elephant shark *Callorhinchus milii* (XM_042344189.1). Rar sequences belong to the NR1B gene family while Rxr sequences belong to the NR2B gene family for which a robust phylogeny allows easy recognition of the shark sequences from their similarity to the *Rhincodon typus* identified sequences (Fonseca et al., 2020).

A set of forward and reverse primers were designed to amplify selected genes from cDNAs obtained from a mix of jaw (48 mm long embryo) and tail (28 mm long embryo). *aldh1a2, rara, rarb, rxrb* and *rxrg* were strongly amplified, *cyp26a1, cyp26b1* and *cyp26c1* were weakly amplified, but *aldh1a1, aldh1a3, rxra* and *rarg* could not be amplified from our jaw/tail cDNA library. We also amplified selected sequences from our in-house cDNA library (Table S1) in the case of *aldh1a2, cyp26a1, cyp26b1* and *cyp26c1*. All PCR products were then ligated in pGEM-T easy vector following a TA cloning kit protocol (Promega, Madison, WI, USA) to generate a matrix for RNA probe synthesis, except for *cyp26b1* and *aldh1a2* which were kept ligated in pSPORT1 as the original cDNA library clone.

## Small-spotted catshark probes and *in situ* hybridization

Antisense RNA digoxigenin-UTP probes were transcribed using SP6 or T7 RNA polymerases (Roche, Basel, Switzerland), according to the orientation of the insert in the plasmid pGEM-T Easy, or pSPORT1 for *cyp26b1* and *aldh1a2*. Whole mount i*n situ* hybridizations were performed on small-spotted catshark lower jaws of stage 32 embryos (body length ranging from 40 to 55 mm) and tails of stage 29 embryos (body length ranging from 25 to 30 mm) to observe gene expression at the four characteristic developmental stages of tooth and scale (Reif, 1980; Debiais-Thibaud et al., 2011; Debiais-Thibaud et al., 2015). For each gene, i*n situ* hybridizations were done in parallel with positive control (earlier embryo with restricted expression pattern, see Fig. S1). *In situ* hybridizations were performed as previously described with samples treated with proteinase K (10 µg/ml) (Freitas & Cohn, 2004; Debiais-Thibaud et al., 2011). Samples were post-fixed in 4% PFA after whole mount *in situ* hybridization, then cleared and stored in glycerol at 4 °C until photographed.

## Histological sectioning

Whole jaws and tails either after *in situ* hybridization or at the end of treatments were transferred to absolute ethanol, then in butanol and finally embedded in paraplast for 15 µm cross-sections, and were cut along the antero-posterior and dorso-ventral axes, respectively.

## Pharmacological treatments

Pharmacological treatments were performed independently on eleven batches of embryos (A to K, see Table S2). Half of each batch was treated either with RA (all-trans retinoic acid, Sigma) or with a retinaldehyde dehydrogenase reversible competitive inhibitor, DEAB (4-Diethylaminobenzaldehyde; Sigma, St. Louis, MO, USA) (*Russo, Hauquitz & Hilton, 1988*), while the other half was incubated under the corresponding control conditions. Different batches were used to replicate the experiments (for example, RA treatment on teeth was made three times with individuals from batches G–I). Embryos were incubated in the dark at 16 °C. Various RA and DEAB concentrations were diluted in sea water filtered at 0.2 µm with antibiotic and antifungal (pen-strep 1X; Sigma, St. Louis, MO, USA) from $10^{-2}$M and $10^{-1}$M stock solutions in DMSO, respectively. Control embryos were treated with equivalent concentrations of DMSO. The water was continuously oxygenated using a pump. To allow drug access into the embryo, an opening was made on both sides of the egg case before the start of the experiment.

Based on the data available in the literature for zebrafish (*Kopinke et al., 2006*; *Laue et al., 2008*; *Spoorendonk et al., 2008*; *Gibert et al., 2010*), preliminary experiments tested the survival and phenotypic changes of the embryos as a function of the duration of exposure to the drug, its concentration, and the frequency of bath renewal (Table 1). We selected the protocols with observable phenotypic effects and the low mortality for further studies (bold in Table 1).

For the caudal primary scale development study, we exposed small-spotted catshark stage 28 embryos (batches A–F) given that the first buds develop at stage 29 (*Mellinger & Wrisez, 1993*; *Debiais-Thibaud et al., 2011*). Each drug experiment ended after one week, expected to be the time needed for the morphogenesis of the first scales to be complete (*Debiais-Thibaud et al., 2011*). Exposure to RA lasted either 48 or 72 h, with baths changed daily, and treated embryos were maintained in sea water with pen-strep (baths changed daily) for the remaining 5 or 4 days. DEAB was applied continuously over 5 days (baths changed daily), and embryos were maintained in the last bath for the remaining 48 h. At the end of the experiments, embryos were euthanized, measured and caudal primary scales buds were counted under a binocular magnifying glass before fixation and dehydration for storage. Usually the buds appear in pairs, the number on the right being equal to that on the left. In case of asymmetry (frequently observed in normal development), the side with the higher number of buds was considered. To identify stages of scale morphogenesis, tails of embryos treated with either 48 h RA $10^{-6}$M (batch B) or DEAB $10^{-4}$M (batch D) were serially sectioned in a transverse plane for histological staining as described above (raw data in Table S3). Histological sections were

Table 1 Assessment of the protocols carried out for pharmacological treatments.

| Drugs | Embryo stages | Duration of drug application | Final concentrations | Frequency of baths changes | Survival rate | Statistical analysis | | |
|---|---|---|---|---|---|---|---|---|
| | | | | | | $p$-value (Gauss) | $p$-value (Poisson) | $p$-value (Q-Poisson) |
| DEAB | 28 | 1 week | $10^{-4}$M/0.1% DMSO | Every 48 h | 100% ($n = 4$) | | 0.799 | |
| DEAB | 28 | 1 week | $5 \times 10^{-5}$M/0.05% DMSO | Every 48 h | 100% ($n = 12$) | | 0.631 | |
| DEAB | 28 | 1 week | $10^{-5}$M/0.05% DMSO | Every 48 h | 92% ($n = 12$) | | 0.953 | |
| DEAB | 28 | 1 week | $10^{-3}$M/1% DMSO | Daily | 0% ($n = 6$) | | ND | |
| **DEAB** | **28** | **1 week** | **$10^{-4}$M/0.1% DMSO** | **Daily** | **68% ($n = 41$)** | | **0.269** | |
| DEAB | 31–32 | 3 weeks | $10^{-4}$M/0.1% DMSO | Daily | 0% ($n = 9$) | ND | | |
| DEAB | 31–32 | 3 weeks | $10^{-5}$M/0.01% DMSO | Daily | 10% ($n = 29$) | ND | | |
| DEAB | 31–32 | 3 weeks | $10^{-4}$M/0.1% DMSO | Every 48 h | 0% ($n = 8$) | ND | | |
| DEAB | 31–32 | 3 weeks | $5 \times 10^{-5}$M/0.05% DMSO | Every 48 h | 12% ($n = 8$) | ND | | |
| DEAB | 31–32 | 3 weeks | $10^{-5}$M/0.01% DMSO | Every 48 h | 37% ($n = 8$) | ND | | |
| **DEAB** | **31–32** | **1 week** | **$5 \times 10^{-5}$M/0.05% DMSO** | **Daily** | **56% ($n = 34$)** | **2.826 e$^{-10}$** | | |
| **DEAB** | **31–32** | **1 week** | **$10^{-5}$M/0.01% DMSO** | **Daily** | **70% ($n = 30$)** | **0.22** | | |
| **RA** | **28** | **48 h** | **$10^{-6}$M/0.01% DMSO** | **Daily** | **94% ($n = 31$)** | | | **0.0015** |
| RA | 28 | 48 h | $10^{-7}$M/0.01% DMSO | Daily | 100% ($n = 8$) | | 0.015 | |
| RA | 28 | 48 h | $10^{-8}$M/0.01% DMSO | Daily | 100% ($n = 8$) | | | 2.232 e$^{-5}$ |
| **RA** | **28** | **72 h** | **$10^{-6}$M/0.01% DMSO** | **Daily** | **100% ($n = 14$)** | | | **0.105** |
| **RA** | **31–32** | **1 week** | **$10^{-6}$M/0.01% DMSO** | **Daily** | **58% ($n = 33$)** | **1.664 e$^{-9}$** | | |

Note:
Small-spotted catshark embryos were treated at stage 28 for scale development and stage late 31/early 32 for tooth development. Different treatment times, drug concentrations and bath change frequencies were tested. The $p$-values and the regression model used to evaluate the pharmacological effect on scale or tooth number are indicated. Significant $p$-values are indicated with an underline. We selected protocols that induced a morphological effect with the lower mortality effect (indicated in bold). An exposure to DEAB $10^{-4}$M was selected for stage 28 embryos even though no significative effect could be observed, a higher dose ($10^{-3}$) being lethal. For stage 31/32 embryos, continuous exposure to DEAB $10^{-5}$ or $5 \times 10^{-5}$M for a week was selected, other protocols providing low survival rates. The dose of $10^{-6}$M and a short exposure to RA were selected for both 28 and 31/32 stages with acceptable survival rates and with a significant effect. Q-Poisson, Quasi-Poisson; ND, not determined; $n$, number of treated embryos.

counterstained with Mayer's hematoxylin solution (1 g/l; Sigma, St. Louis, MO, USA) and eosin Y solution (0.5% aqueous Sigma; Sigma, St. Louis, MO, USA).

For the tooth development study, small-spotted catshark embryos (batches G–K) were exposed at late stage 31 or beginning of stage 32 (*Ferreiro-Galve et al., 2010*) before the appearance of first tooth buds (*Debiais-Thibaud et al., 2015*). Experiments were ended after 3 weeks, which is the estimated time necessary for all stages of morphogenesis to occur in the first developing teeth (*Debiais-Thibaud et al., 2015*). Exposure to pharmacological agents (RA $10^{-6}$M or DEAB at $10^{-5}$ and $5 \times 10^{-5}$M) was performed for 5 days with daily changes of the respective medium. The embryos were then maintained in the last bath for the remaining 48 h and placed in sea water with pen-strep (baths changed every 48 h) for the next 2 weeks. Embryos were then euthanized, measured and fixed.

Because tooth buds were not visible under a stereo microscope, a total of 104 whole lower jaws (*i.e.*, both jaw halves) were dissected and serially sectioned sagittally for histological staining and counting of tooth buds. For each tooth or caudal primary scale bud, we observed all the stages of odontode morphogenesis: early morphogenesis (EM), late morphogenesis (LM), early differentiation (ED) and late differentiation (LD) (*Borday-Birraux et al., 2006*; *Debiais-Thibaud et al., 2007*; *Debiais-Thibaud et al., 2011*). Only the odontode buds of the LM to LD stages were counted, because the EM stage is more difficult to observe and may become a potential source of error (raw data in Table S3).

The number of buds at different developmental stages (LM, ED and LD) was compared in treated embryos and their respective controls (embryos from all successive batches exposed to the same treatment were pooled, although only the batch D was used to characterize the DEAB exposure on scales, see Table S3). The caudal primary scale buds on both the right and left sides were considered. Tooth buds were counted on the whole lower jaws.

## Statistical analyses

For the pharmacological treatment experiments, only live embryos at the end of the experiment were analyzed and the buds counted (scale or tooth). Statistical tests were performed with the software R version 4.0.2 (https://www.r-project.org). Effects of pharmacological treatments on odontodes and on embryo body length were tested using a generalized linear model (glm function) considering batch experiments and the interaction between treatment and batch. Gaussian regression model was applied for data of tooth buds and embryo body length while Poisson regression model was applied for caudal primary scale buds due to the skewed distribution toward null values (quasi-Poisson when the data were very scattered). Influence of explanatory variables (treatment and batch) and of interaction was tested using type 2 Anova ("Anova" function from "car" library) and the models were simplified when interaction and batch effect did not significantly improve the model.

The effects of the RA and DEAB treatments on the stages of tooth and scale morphogenesis between the treated and their respective controls were tested using the Mann-Whitney Wilcoxon test ("wilcox.test" function). This was also the case for testing the homogeneity of the developmental scores between treatments (RA or DEAB) and their respective DMSO controls.

## Developmental criteria in stage 32 embryos

Embryos chosen for the tooth development study were staged 'late stage 31/early stage 32': the transition between these stages is not sharp, and stage 32 is a very long stage in the developmental table of the small-spotted catshark (50 days in *Ballard, Mellinger & Lechenault, 1993*; *Ferreiro-Galve et al., 2010*) so that the embryos at the end of the experiments were still at stage 32. As little data is available in the literature to finely describe stage 32, we determined developmental criteria from the observation of reference specimens whose body length ranged from 40 to 58 mm, with steps of 2 to 3 mm (*n* = 5 for each size). The identified developmental criteria are presented in Figs. S2A and S2B. They

**Table 2 Final scores of reference embryos.**

| Reference embryo body length (mm) | 40 | 43 | 45 | 48 | 50 | 52 | 55 | 58 |
|---|---|---|---|---|---|---|---|---|
| Average of final scores | 0.80 | 1 | 2 | 2.6 | 7.2 | 10 | 12.5 | 17.8 |
| SD | 0.45 | 0 | 1.22 | 1.14 | 4.09 | 2.45 | 1.34 | 1.10 |

Note:
   For each reference embryo body length, we observed five specimens. SD, standard deviation.

focus on the pigmentation pattern of the top of the head, back, dorsal and caudal fins, front of the eye and retina. For the pigmented retina, its shape was also considered. We also considered the length of the gill filaments. For each criterion except the gills, a score was assigned based on pigmentation level (see Fig. S2C).

Gill filament length (g) is dynamic; it peaks at stage 31 and then regresses until filaments completely disappear during stage 32 (*Ballard, Mellinger & Lechenault, 1993*; *Musa, Czachur & Shiels, 2018*). We assigned a score of 0 when the gill filaments were at their maximum extent (filaments length is about half the distance from the tip of the snout to the pectoral fin base), a score of 1 when they partially regressed (filament length is about one third the distance from the tip of the snout to the pectoral fins) and a score of 2 when gill filaments were almost completely gone.

The scores for each criterion were summed, giving a total score for each embryo. For a given body length, the total scores of the five studied reference embryos were averaged (see Table 2).

No such developmental table could be established for the younger experimental cohort (caudal primary scale development) because of the short experimental time (only 1 week).

# RESULTS

## Genes coding for proteins of the RA signalling pathway are expressed in developing teeth and caudal primary scales in the small-spotted catshark

In our locally available cDNA library and the publicly available genomic sequence of the small-spotted catshark, we identified putative coding sequences for genes from the RA signalling pathway: three genes encoding the RA-synthesizing enzymes Raldh (*aldh1a1, aldh1a2 and aldh1a3*), three genes encoding the catalytic enzymes Cyp26 (*cyp26a1, cyp26b1 and cyp26c1*), three genes encoding Rar receptors (*rara, rarb and rarg*) and three genes encoding Rxr receptors (*rxra, rxrb and rxrg*). Of these twelve genes, *rarg, rxra, aldh1a1* and *aldh1a3* could not be amplified from cDNAs generated from RNA extracts of embryonic small-spotted catshark jaw or tail, suggesting these genes are poorly transcribed in these areas of the embryo (not shown).

The expression of the remaining eight genes was investigated by *in situ* hybridization during the development of the first teeth and caudal primary scales on lower jaws of 40 to 55 mm long small-spotted catshark embryos (stage 32), and dissected tails of 25 to 30 mm long embryos (stage 29) respectively (*Ballard, Mellinger & Lechenault, 1993*; *Ferreiro-Galve et al., 2010*).

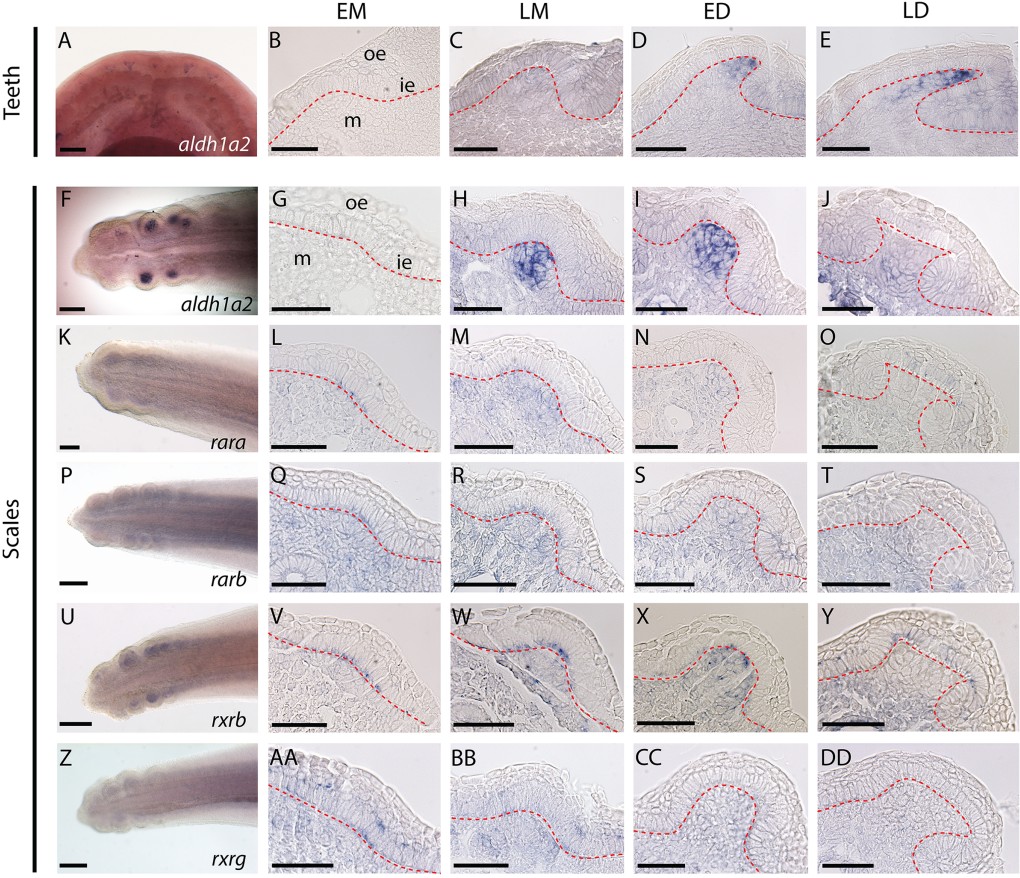

**Figure 1 Gene expression patterns during tooth and caudal primary scale development.** (A–E) Whole mount dorsal view of lower jaw of a 50 mm long embryo. (AA–AD) Longitudinal sections of jaws (45–50 mm long embryos) showing tissue specific expression in tooth bud from EM to LD stages. (G–J, L–O, Q–T, V–Y, AA–DD) Whole mount hybridized tails of 25–28 mm long embryos with the gene probe shown in the images; anterior is to the right, dorsal is to the top. (G–J, L–O, Q–T, V–Y, AA–DD) Transversal sections of hybridized tails showing tissue specific expression of genes in scale bud from EM to LD stages. EM, early morphogenesis—thickening of the odontogenic epithelium (ie) and condensation of the underlying mesenchyme (m); LM, late morphogenesis—bell-shaped bud; ED, early cytodifferentiation—constriction at the base of the bud; LD, late cytodifferentiation—matrix deposition; ie, inner epithelium; m, mesenchyme; oe, outer epithelium. Scale bars: (A, F, K, P, U, Z) 200 μm; (B–E, G–J, L–O, Q–T, V–Y, AA–DD) 50 μm. The red dotted lines indicate the boundary between epithelium and mesenchyme.

Only *aldh1a2* was expressed during the development of both teeth and caudal primary scales (Figs. 1A and 1F). Low levels of expression of *rara*, *rarb*, *rxrb* and *rxrg* were detected in developing caudal primary scales (Figs. 1K, 1P, 1U and 1Z). No expression of the three *cyp26* genes was detected in either tooth or scale buds, although their transcripts were weakly amplified from our jaw/tail cDNA library (Fig. S3 and see material and methods section).

Expression of *aldh1a2* was strong in the mesenchyme of both tooth and scale buds (Figs. 1B–1E and 1G–1J, respectively), with an apparent increasing intensity from late morphogenesis (LM) to late differentiation (LD) stage for tooth buds and a maximum intensity at early differentiation (ED) stage for scale buds. Expression of *rara, rarb* and *rxrg*

was detected at low intensity in the mesenchyme of the tail, including buds from EM to ED stages. At the LD stage, *rara*, *rarb* and *rxrg* expression were turned off in the scale buds but remained weakly expressed in the rest of the mesenchyme (Figs. 1L–1O, 1Q–1T and 1AA–1DD, respectively). Their transient expression was also observed in cells from the odontogenic epithelium during EM and LM. The expression of *rara* was detected in the superficial odontogenic epithelium of the scale buds at LD (Fig. 1O). Finally, *rxrb* was expressed in the epithelial cells from EM to LD stages and in the mesenchyme at LM and ED stages (Figs. 1V–1Y).

## Pharmacological inhibition of RA synthesis affects tooth and caudal primary scale development

As *aldh1a2* expression is highly induced in both tooth and caudal primary scale development, inhibition of Raldh2 activity is expected to modify the development of small-spotted catshark odontodes. Stage 28 embryos (batches D to F) and late stage 31/early 32 embryos (batches J and K), were exposed to DEAB at $10^{-4}$ M (scale exposure), and $10^{-5}$ M or $5 \times 10^{-5}$ M (tooth exposure), respectively, continuously for one week (see Table 1 and Table S2). Scale (Fig. 2A) and tooth (Fig. 2C) bud counts revealed no significant difference between control and DEAB exposure at $10^{-4}$ M and $10^{-5}$ M conditions, respectively. In contrast, the number of tooth buds was reduced by one-third after exposure to $5 \times 10^{-5}$ M DEAB compared to controls, where a mean of 30 tooth buds were observed (Fig. 2E).

We then investigated how DEAB can disrupt scale and tooth morphogenesis by observing the histological phenotype. No significant difference could be observed between the treated and control embryos in the number of scale bud for each developmental stage (Fig. 2B), nor in the DEAB $10^{-5}$ M exposure during tooth development (Fig. 2D). In contrast, the $5 \times 10^{-5}$ M DEAB-treated embryos showed a significantly lower number of teeth in each developmental stage, with a stronger difference in the LD stage (Fig. 2F, *p*-values 0.03, 0.01 and 0.0004 for the comparison of mean values at LM, ED and LD stages respectively).

## Exogenous RA treatment inhibits the development of teeth and caudal primary scales

As expression of *rara*, *rarb*, *rxrb* and *rxrg* was detected in scale development (Figs. 1C–1F), exogenous RA is expected to modify the morphogenesis of small-spotted catshark odontodes. Stage 28 embryos (batches A to C) were exposed to $10^{-6}$ M RA for either 48 or 72 h and late stage 31/early 32 embryos (batches G to I) were exposed to $10^{-6}$ M RA for one week (see Table 1 and Table S2). A significant effect of RA treatment could be detected on scale bud number when considering stage 28 embryos treated for 48 h (*p*-value = 0.038 when all batches are considered). However, batch C48 behaves differently from the two other ones, most probably due to the smaller length of control embryos compared to the treated ones (see Fig. S4). As a result, we considered only batches A and B in subsequent analyses (Fig. 2 and Table 1). The exposed embryos showed on average only one scale bud, compared to four in control embryos (Fig. 2G, *p*-value = 0.0015). A decreasing trend in the

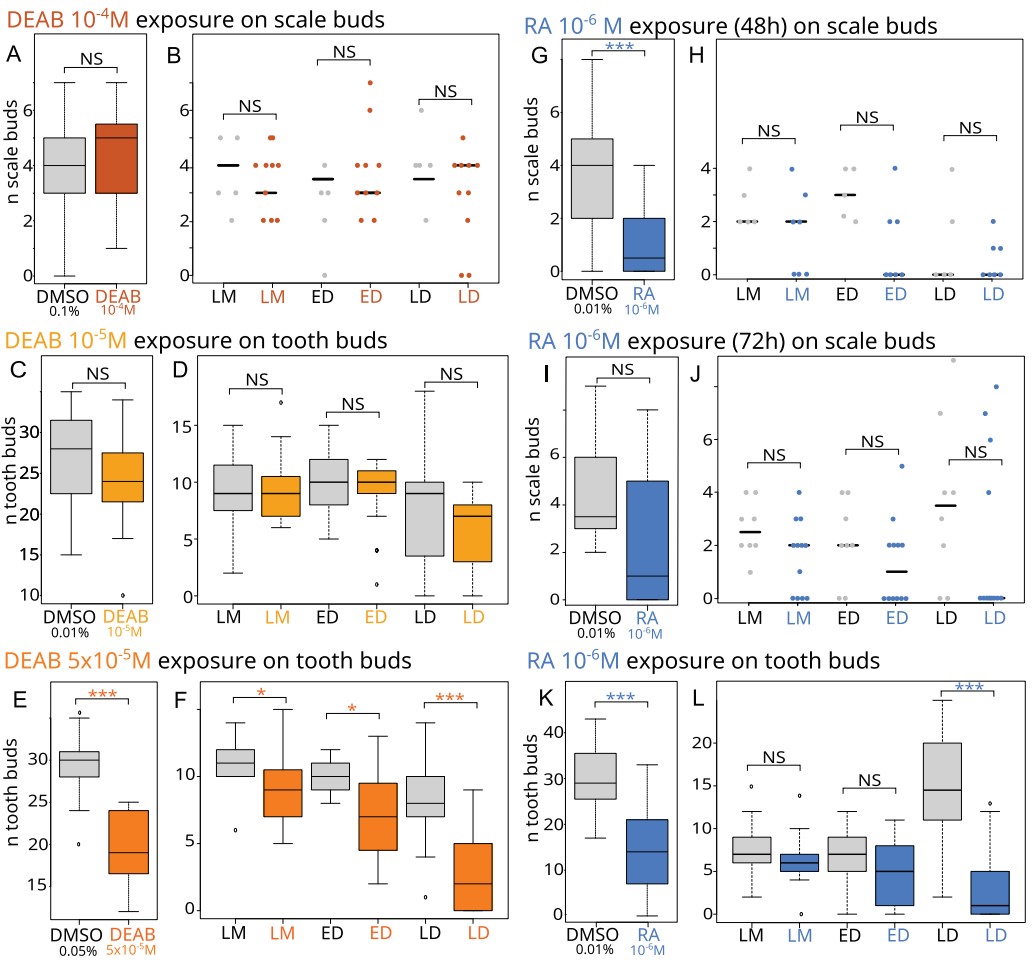

**Figure 2 Effects of RA synthesis inhibition with DEAB or exogenous RA treatments on ondotodes development and morphogenesis.** (A, C, E, G, I, K) Box plots representing the effects of DEAB (A, C, E) or RA (G, I, K) on scale (A, G, I) and tooth (C, E, K) bud number compared to their respective DMSO control conditions. n represents the number of embryos for which a number of tooth or scale buds could be determined (for details, see Table S2). In (G), only batches A and B were considered. (B, D, F, H, J, L) Stripcharts (B, H, J) or box plots (D, F, L) representing the number of scale or tooth buds respectively in the three stages of scale morphogenesis, LM, ED and LD in specimens showing scale or tooth buds after DEAB (B, D, F) or RA (H, J, L) treatments compared to DMSO controls. In this analysis, the dorsal scale buds on the right and left sides were considered. The representation in stripcharts was favored for the scales given the small number of embryos analyzed and the dispersion of the data. For boxplots and details of the number of samples analyzed, see Table S3. The grey color represents the control conditions, the orange colors the DEAB treatments (the darker the color the higher DEAB concentration), the blue color the RA treatment. In both type of diagrams, the black line represents the median value, * for *p*-values < 0.05 and *** for *p*-values < 0.001. In boxplot representation, box height depicts first and third quartiles, whiskers represent extreme values.

number of scale buds was also observed in the case of a 72-h exposure to RA (batch C72, Fig. 2I) although results are not statistically significant (see Table 1 and Table S2), probably due to the dispersion of values and the low number of embryos. For tooth development, the number of tooth buds after RA treatment was reduced by 50%, to about 15 teeth compared to an average of 30 in the controls (Fig. 2K, *p*-value = 1.664 $e^{-9}$). As a consequence, RA inhibits tooth initiation during the time of treatment.
We then investigated how RA treatments can disrupt scale and tooth morphogenesis by observing the histological phenotype. The analysis of scale morphogenesis was performed on embryos treated for 72 h and on embryos only from batch B for the 48-h treatment (see Table S3). In RA-treated embryos in which scale buds are observed, no significant difference in the mean number of scales at each developmental stage is observed (Figs. 2H and 2J) compared to control. In contrast, the mean number of tooth buds in the late differentiation stages was significantly lower in treated embryos than in control embryos ($p$-value = 9.436 $e^{-6}$, Fig. 2L).

## Exogenous RA has tooth-specific effect while DEAB effect is due to a general developmental delay

We observed a reduction in the number of teeth and a delay in tooth morphogenesis in embryos treated with RA or DEAB as compared to controls. As the treated embryos were generally smaller in body length than the control embryos ($p$-value = 0.001 for RA, $p$-value = 1.212 $e^{-11}$ for DEAB $5 \times 10^{-5}$M, but $p$-value = 0.219 for DEAB $10^{-5}$M; see Table S4), we analyzed these embryos for a general developmental delay by examination of several developmental criteria (see Fig. S2).

The relationships between the developmental score and embryo body length are presented in Fig. 3. The embryo length was between 44 and 58 mm in all controls. A total of 91% of the control embryos in the RA treatments are greater than or equal to 49 mm in body length; this proportion is 60% and 78% for the control embryos in the DEAB treatments at $10^{-5}$ and $5 \times 10^{-5}$M, respectively (Fig. 3 and Table S4). The associated total score values ranged from 5 to 20 for all controls, with 94% having a score greater than or equal to 7, which we therefore consider as the lower limit for reference embryos.

In RA-treated embryos, embryo body length was distributed between 41 to 56 mm, with half of the embryos being shorter than 49 mm. However, developmental scores were mostly greater than 7 (82%), making the treated and control animals not significantly different ($p$-value = 0.197) (Fig. 3A and Table S4). As a result, RA treatments did not delay the general development of treated embryos although body length was lower.
The reduction in the number of tooth buds in the treated group appears independent from the developmental score (Fig. 4A).

In $10^{-5}$M DEAB treatments, more than half of the embryos was shorter than 49 mm, but the associated total score values were mostly greater than 7 (84%). Thus, the treated and control populations were not significantly different ($p$-value = 0.507) (Fig. 3B and see Table S4 for details). As a consequence, control and treated embryos show similar relationship between tooth bud number and total score values (Fig. 4B). In contrast, all $5 \times 10^{-5}$M DEAB-treated embryos were significantly shorter than or equal to 49 mm and 1/3 of them had an associated total score lower than 7 ($p$-value = 0.0014) (Fig. 3C).
The reduced number of tooth buds observed in the DEAB $5 \times 10^{-5}$M treated group is then probably due to a more general developmental delay effect (Fig. 4C). We conclude that the $5 \times 10^{-5}$M dose of DEAB results in an overall developmental delay compared to controls which is not observed in the lower dose.

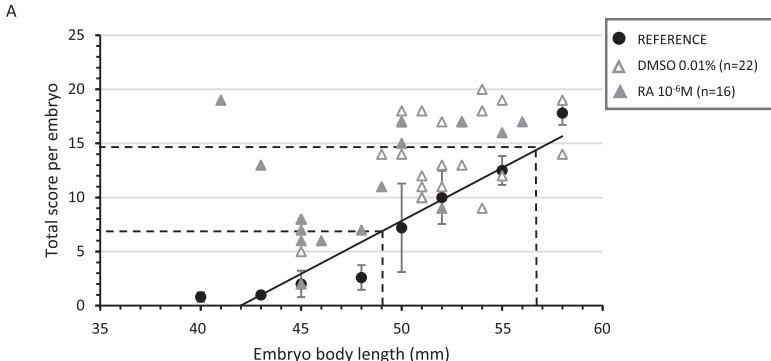

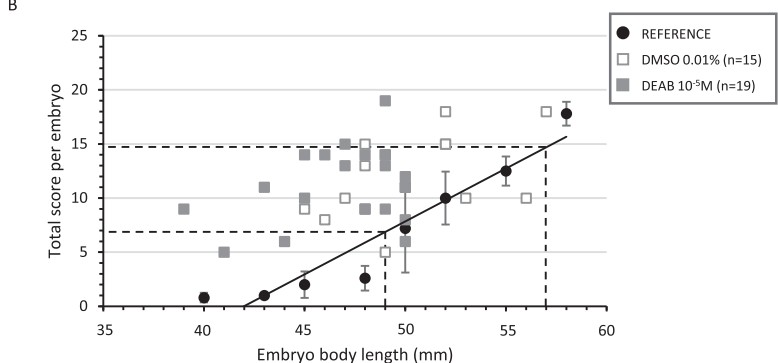

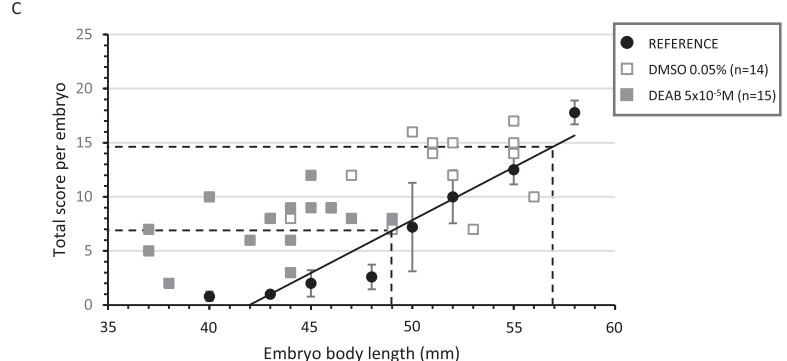

**Figure 3 Correlation between the total score and the body length of treated embryos and their respective controls.** Diagrams representing the distribution of the total score as a function of the corresponding body length for each treated and control embryo in RA $10^{-6}$M (A), DEAB $10^{-5}$M (B) and DEAB $5 \times 10^{-5}$M (C) treatments. The black circles represent the average of the total scores of the five reference embryos for a given length and the black line the trend line. The dotted lines represent the lower and upper limits of the corresponding length and total score expected after treatment (middle 32 stage).

## DISCUSSION

### The RA signalling pathway toolkit is partially present in zones of tooth and caudal primary scale development in the small-spotted catshark

All expected members of the vertebrate RA-signalling toolkit were identified in the genomic and transcriptomic data of the small-spotted catshark (all six genes coding for

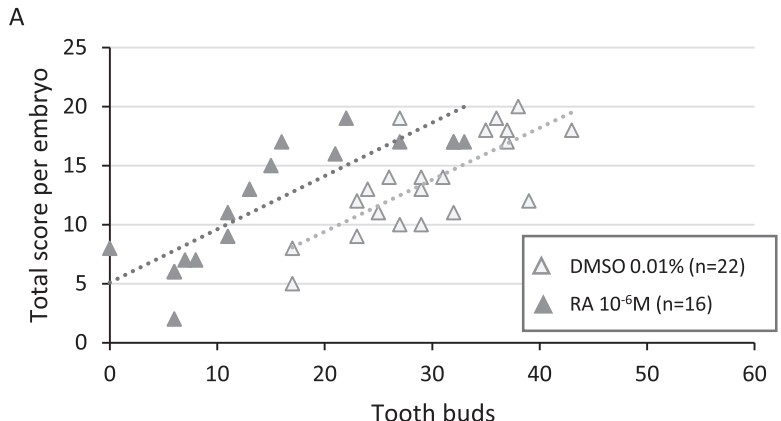

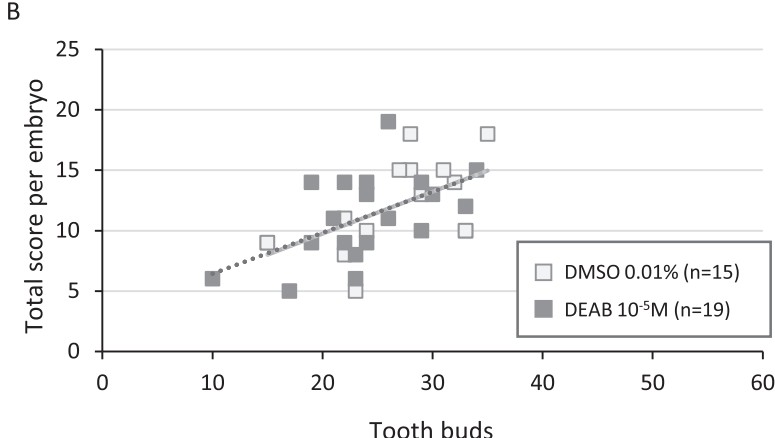

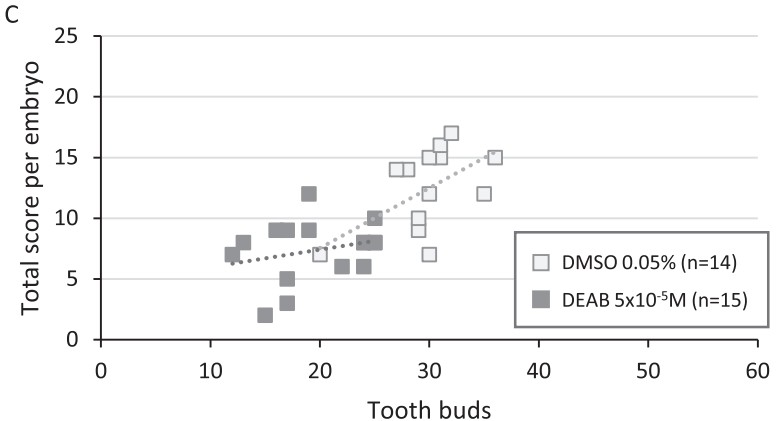

**Figure 4 Correlation between total score and number of tooth buds in treated embryos and their respective controls.** Diagrams representing the distribution of total score as a function of the corresponding number of tooth buds for each treated and control embryo in RA $10^{-6}$M (A), DEAB $10^{-5}$M (B) and DEAB $5 \times 10^{-5}$M (C) treatments. Dotted lines and the light grey line in B represent the trend lines.

both synthesis and degradation enzymes, *aldh1a1* to *3* and *cyp26a1* to *c1*; all six nuclear receptors *rara* to *g* and *rxra* to *g*). Among these genes, only *aldh1a2* expression was detected in developing tooth buds by *in situ* hybridization, while *aldh1a2* together with *rara*, *rarb*, *rxrb* and *rxrg* expression was detected in developing scale buds. Our results show that the RA signalling pathway is present in caudal primary scale buds, similar to what has been described in mouse tooth development (*Bloch-Zupan et al., 1994*; *Sasaki et al., 2010*). This is also the case during the development of the first zebrafish pharyngeal tooth, where the expression of *aldh1a2* and *rar* genes, although not strictly observed in dental cells, was detected in the pharyngeal region (*Hale et al., 2006*; *Gibert et al., 2015*). On the other hand, the lack of RA nuclear receptor expression in tooth buds questions the function of this signalling pathway in tooth development. However, expression levels of retinoic acid receptors might be low (under the detection threshold), yet still be active. The RA signalling pathway might then be active during tooth development (as suggested by the steady expression of *aldh1a2* in the mesenchyme of tooth buds) despite the lack of detectable expression of nuclear receptors in the surrounding cells. No expression of the *cyp26* genes was detected in either teeth or caudal primary scales, suggesting that expression of *cyp26* genes would be below the detection threshold or that RA-degrading enzymes are not active during tooth and scale bud development in the small-spotted catshark. These results are in contrast to what has been observed in zebrafish teeth where *cyp26b1* is highly expressed in the mesenchyme of tooth buds during morphogenesis (*Gibert et al., 2015*).

In this study we focussed on the development of the first teeth and of the primary caudal scales, both being the first components of the dentition and dermal denticles respectively. Further descriptions in later developing tooth germs and other sets of dermal denticles that develop later in ontogenesis are needed to generalize our observations to all components. Most genes previously studied in the small-spotted catshark were shown to be expressed similarly in tooth and scale buds despite few exceptions (*Debiais-Thibaud et al., 2015*; *Berio & Debiais-Thibaud, 2021*; *Cooper et al., 2023*). As a consequence, a wider view in the small-spotted catshark might help further interpret the observed differences in comparison to the teeth of the mouse, zebrafish or medaka where only one type of odontode was surveyed. Our focus on first developing teeth and scales allowed us to further test how interference with the RA signalling pathway modifies their development at early, experimentally controllable, stages.

## Tooth and caudal primary scale initiation is altered by RA signalling disruption in the small-spotted catshark

RA exposure led to an arrest in scale and tooth bud initiation: this effect was detected on scales with the doses ranging from $10^{-6}$ to $10^{-8}$M (Table 1). We focused on the effect of $10^{-6}$M dose: younger embryos exposed for 2 days (out of 7 days of experimental growth) possessed only one fourth of the control scale number, while older embryos exposed for 5 to 7 days (out of 21 days of experimental growth) possessed half of the control number of tooth buds. As development of successive tooth/scale buds is a temporally continuous process in shark, a direct effect of RA is expected to be proportional to the duration of

exposure (*e.g.*, 5 to 7-day exposure out of 21 should have led to only 24% to 30% loss of the tooth buds, see Fig. 2). However, the inhibitory effect on tooth development was much greater with a 50% reduction in tooth buds compared to controls. Over-exposure to RA therefore blocked tooth initiation not only over the exposure period, but probably also had a delayed effect on the following recovery to normal rate of tooth bud initiation. The missing teeth in treated embryos are teeth at stage LD in control embryos, suggesting our treatment has an effect only on the initiation stage of teeth growing during the time of treatment (the founder teeth), and not on later development processes: *i.e.*, the morphogenesis of the founder teeth or the initiation of subsequently developing tooth buds. Our experimental set-up involves that the treatment happens only in the initiation of the founder teeth: as a result, it cannot discriminate if the inhibitory action of RA is specific to these founder teeth or might be general to all subsequent teeth. Further studies interfering with RA signalling at later stages are necessary to evaluate the sensitivity of later developing teeth in the small-spotted catshark. An equivalent evaluation of effect on scale developmental stages is not possible: embryos at the end of the one-week treatment either have no caudal primary scales (Fig. 2G and Table S2) or have a number of scale buds at stages similar to controls (Fig. 2H and Table S3). In this case, the time of recovery is probably too short to reinitiate the process of scale initiation.

RA over-exposure was previously shown to generate additional tooth buds in the zebrafish, supposed to be a result of the modification of neural crest cell patterning (*Seritrakul et al., 2012*) despite other authors suggested RA signalling may impact zebrafish tooth initiation by modification of the endoderm-ectoderm boundary (*Huysseune, Cerny & Witten, 2022*). In the small-spotted catshark, very little is known about how and when neural crest cells associate to the skin and dental lamina (*Gillis, Alsema & Criswell, 2017*), or how the epithelial compartment of either scales or teeth is patterned (*Huysseune, Cerny & Witten, 2022*). However, our data are congruent with an effect of RA over-exposure on the tooth and scale initiation process as the product of impaired epithelial-mesenchymal molecular interactions. These may be associated to impaired signalling by neural crest-derived cells located in the mesenchymal compartment at the initiation site, as it was shown for the dental germ in mammals (*Lumsden, 1988*) and for the feather patterning and development in birds (*Chuong et al., 1992*; *Chuong, 1993*).

On the other hand, pharmacological exposure to DEAB is considered inhibitory to RA signalling, due to the inhibition of Raldh enzymes (*Morgan et al., 2015*). The various exposure protocols we tested here showed high mortality effects, except for the $10^{-5}$M/0.01% DMSO (70% survival) and $5 \times 10^{-5}$M (56% survival) protocols. In the DEAB treatments, we showed no significant modification in the number of scale or tooth buds, except for the $5 \times 10^{-5}$M treatment, where the number of tooth buds decreased by 30% compared to the controls with lower number of teeth in each developmental stage. However, the $5 \times 10^{-5}$M treatment was associated with an overall delay in development. We evaluated this developmental delay by the characterization of several phenotypical cues that involve pigmentation, retina shape and gill filament length, despite the fact that pigmented cell differentiation may be sensitive to RA-signalling, as shown in metamorphic flounder larvae (*Chen et al., 2020*), but no sensitivity was detected in zebrafish after 48 hpf

(*Li et al., 2010*). Considering the relationship between our developmental score and body length, we considered this score still able to detect global developmental delay rather than simply a putative effect of DEAB on the pigmentation. As a consequence, our results point to a non-specific and general effect of DEAB exposure on the growth processes, that interacts with tooth development. This general effect is congruent with the ubiquitous activity of Raldh in early embryonic development (*Niederreither et al., 2002*). This general effect may erase other, tooth- and scale-specific, aspects of RA signalling as differentiation of odontoblasts and ameloblasts was shown to be affected by high levels of RA signaling (*Wang et al., 2020*; *Morkmued et al., 2016*).

## Comparative perspectives

As a conclusion, higher than endogenous RA signalling appears to be inhibitory for tooth and caudal primary scale initiation in the small-spotted catshark, which is in opposition to previous data gathered in the mouse and zebrafish models (*Kronmiller, Nguyen & Berndt, 1995*; *Gibert et al., 2010*). Several differences in the experimental conditions used to interfere with RA-signalling may be confounding the comparison between species. Tests in the zebrafish were made through phenotyping adults of a mutant line with impaired cyp26 activity, or with a very long treatment period covering the end of the embryonic development and juvenile stages until 12 dph (*Gibert et al., 2015*). These previous results therefore involved both very long-time window for the interference and very long-term effects on the phenotypes: their observation is that half of the cohort showed one additional tooth. The mouse phenotypes were obtained by culturing jaw explants in a medium containing 0.05 μg/mL RA (*Kronmiller, Nguyen & Berndt, 1995*) implying a very modified physical microenvironment for tooth development. If the main conclusion was that additional teeth grew in the diastema, the authors also showed that teeth were often missing at the positions where molars grow in controls: this observation may actually be reminiscent of our observations of missing teeth and scales in the small-spotted catshark. In addition to a variety of experimental conditions, the biological contexts for tooth development in these species are highly diverse. Shark teeth are permanently initiated through sequential placode initiation in a dental lamina that includes stem cells (*Fraser et al., 2019*) and that most probably interacts with cranial neural crest cells that populate the underlying head mesenchyme at early stages (*Debiais-Thibaud et al., 2013*). However, mouse supernumerary teeth develop from loci of tooth inhibition (secondary loss of tooth growth in diastema (*Kronmiller, Nguyen & Berndt, 1995*)) where exogenous RA might act as an inhibition leverage. On the other hand, zebrafish supernumerary teeth are pharyngeal and they were interpreted in the light of branchial arch competency for tooth development and anterior endoderm competency (*Seritrakul et al., 2012*; *Huysseune, Cerny & Witten, 2022*). Very different context for tooth initiation in the different lineage of vertebrates might enhance differences in RA signalling function in this process. On the other hand, caudal primary scales develop successively in a posterior to anterior progression, very early and at well-defined locations (*Ballard, Mellinger & Lechenault, 1993*) suggesting patterning by very early migrating trunk neural crest cells, and they are not renewed. This situation is also very different from the shark tooth initiation process

but, we show similar impact of RA signalling disturbance on both organs. RA signalling is therefore involved in odontode initiation in the shark lineage. However, the discrepancies in the function of RA signalling between vertebrate lineages prevent the identification of ancestral aspects, and highlight the versatility of this regulatory system in participating in various aspects of epithelial and mesenchymal patterning.

## ACKNOWLEDGEMENTS

The authors thank the staff of the CNRS animal facility in Gif-sur-Yvette for their care of the small-spotted catshark embryos. We thank Didier Casane for providing the sequence of the *cyp26* and *aldh1a2* genes from the *Scyliorhinus canicula* cDNA library and Silvan Oulion for cloning *aldh1a2* in pSPORT1. We thank the two reviewers for their help in improving the manuscript.

### Funding

The authors received no funding for this work.

### Competing Interests

The authors declare that they have no competing interests.

### Author Contributions

- Isabelle Germon conceived and designed the experiments, performed the experiments, analyzed the data, prepared figures and/or tables, authored or reviewed drafts of the article, and approved the final draft.
- Coralie Delachanal performed the experiments, prepared figures and/or tables, and approved the final draft.
- Florence Mougel analyzed the data, authored or reviewed drafts of the article, and approved the final draft.
- Camille Martinand-Mari performed the experiments, analyzed the data, authored or reviewed drafts of the article, and approved the final draft.
- Mélanie Debiais-Thibaud conceived and designed the experiments, performed the experiments, analyzed the data, prepared figures and/or tables, authored or reviewed drafts of the article, and approved the final draft.
- Véronique Borday-Birraux conceived and designed the experiments, performed the experiments, analyzed the data, prepared figures and/or tables, authored or reviewed drafts of the article, and approved the final draft.

### Animal Ethics

The following information was supplied relating to ethical approvals (*i.e.*, approving body and any reference numbers):

Handling of small-spotted catshark embryos followed all institutional, national, and international guidelines [European Communities Council Directive of September 22, 2010
(2010/63/UE)] and no further approval by an ethics committee was necessary as the biological material is embryonic.

## Data Availability

The raw data of the tested protocols for the pharmacological treatments are available in Table 1.

Additional raw data, including the number of teeth and scale buds after the selected pharmacological treatments, are available in Supplemental Files.

## Supplemental Information

Supplemental information for this article can be found online at http://dx.doi.org/10.7717/peerj.15896#supplemental-information.

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
