# Peer review of "Interference with the retinoic acid signalling pathway inhibits the initiation of teeth and caudal primary scales in the small-spotted catshark Scyliorhinus canicula"

_PeerJ, doi:10.7717/peerj.15896_

## Round 0.1 · original submission · Major Revisions

I understand both reviewers' comments are quite extensive, and am convinced your study will make a significant contribution in the field of retinoid research. I feel that both reviewers raise a number of points that will help clarifying the experimental strategy and the resulting data, and when necessary refine or re-discuss their interpretation. I appreciate, for instance, the comments of the second reviewer who suggests (to a reasonable extent) re-working some parts of the Introduction or Discussion. Indeed this could make the article more attractive, clearer and more appealing to readers non-familiar with this field, and altogether would improve its impact.

I will be glad to receive a revised version of the manuscript at your earliest convenience.

·

Basic reporting

I have entered all my comments in a single document - see point 4.

Experimental design

I have entered all my comments in a single document - see point 4.

Validity of the findings

I have entered all my comments in a single document - see point 4.

Additional comments

The paper by Germon and colleagues is a detailed study of the role of retinoic acid in tooth and caudal scale development in the lesser catshark.

Different from other studies that have examined RA influence, this study tested many concentrations, lengths of treatments and also considered the possibility of a general developmental delay. Such a depth of analysis is not evident given that the embryos are not typical model organisms available in unlimited numbers.

The study strikes me as a very honest and unprejudiced analysis, despite the fact that the results point in a direction that contradicts similar studies on other taxa, and despite the accepted homology between scales and teeth, a viewpoint also supported by some of the authors in previous papers. Their results lead the authors to conclude that the RA signaling pathway must be very versatile and an ancestral pattern cannot be deduced.

I think it is important nevertheless that the authors clarify in more detail how exactly the tooth buds were counted. Given the curvature of the lower jaw, and using sagittal sections, one has to count on serial sections as well as avoid double counting. How was this done? Also, do the numbers refer to one half jaw or the complete jaw (left+right)? More importantly, was only the first formed tooth in the family counted (i.e. were only tooth families counted), or do the numbers also encompass multiple teeth in one family? This is relevant because sequential teeth in a tooth family may not be under the same molecular control as the founder tooth in a family (especially regarding their initiation). In theory, a lower number of teeth could still signify a higher number of tooth families (thus, initiations) but with less or no successor teeth. Thus, total numbers may give a skewed result. Therefore, I think it is important to distinguish between tooth numbers and number of tooth families. Can the authors comment on this (especially in view of the discussion lines 396-399), and explain exactly how the counting was done? Also, does the method used imply that around 100 lower jaws were serially sectioned? (i.e., the three groups on the right on Table S2, as well as in Table S3). If so, this is an amazing piece of work. In Tables S2 and S3, the numbers (n=..) do not actually correspond with the number of lines that they cover. Can the authors clarify? Another point that strikes me is that early morphogenesis tooth buds were not considered (although in Line 213 it is said that EM was identified).

To establish the developmental stage – and therefore a possible developmental delay, the authors used pigmentation patterns and gill length. Both melanocytes and gills may themselves be RA-sensitive and therefore not an independent marker of development. Could the authors comment on this? Can the authors in some (other) way ascertain that DEAB may not have a tooth-specific effect? (lines 335-336).

Lastly, what I am missing a little bit is how the authors discuss their results with what is known on other taxa. Assigning differences to versatility of the pathway is of course a straightforward (if not easy) explanation, but perhaps there can be others (differences in methodology, level of analysis, statistics, resolution, developmental origin of the structures involved, …). There are many aspects that are worth exploring, even briefly. Thus, I suggest the authors reflect a little deeper on what the reasons could be that underlie the deviation of their results with those previously published on other taxa.

Other than that, I have only minor remarks.
- The title word ‘disruption’ could be understood as inhibition, and therefore, could be in conflict with the finding (cf. abstract) that inhibiting RA synthesis does not affect scale development.
- I suggest to replace the word ‘labile’ by ‘versatile’
- Line 79: replace ‘Mutant mice for..’ by ‘Mice mutant for…’
- Line 80: replace ‘exposed with’ by ‘exposed to’
- Line 81: ‘and modified tooth identities’: express more clearly
- Line 135: Rarg: not with capital letter?
- Line 159: ‘with known restricted expression pattern’: any reference for this?
- Line 166: replace ‘incubated’ by (for example) ‘transferred’
- Line 171-172: I do not see well how the eleven batches correspond to Table S2 (there is nowhere a legend to Table S2?). In Table S2 I see six groups, each of which has been divided in two – approx. half for controls, half for treatment - , and batch codes (A-K) co-occur in a single group
- Line 197: replace ‘dissymmetry’ by ‘asymmetry’
- Line 204: at which stage exactly is the first tooth bud formed? It is important to know how much time the pharmaceutical treatments were started before the first tooth anlage.
- Line 211: replace ‘loupe’ by ‘magnifying glass’
- Line 242: I assume that this information was crucial because late stage 31 - early stage 32 was used to start the experiment for the tooth bud counts? Please clarify (confirm or correct)
- Line 243: ‘of’ is missing: ‘specimens of which the body length’
- Line 249: replace ‘regress’ by ‘regresses; ‘until it disappears’: not the length disappears, but the gill filaments do
- Line 379: ‘of’ is missing; ‘lack of detectable expression’
- Lines 388-389: Table 1 does not show tooth numbers; furthermore, the Table indicates that these concentrations were used on stage 28 (i.e., for scale bud counts)
- Line 394: replace ‘proportionate’ by ‘proportional’
- Line 437-438: not very comprehensible; please rephrase
- Line 444-445: are you suggesting that trunk neural crest cells migrate to the sites of caudal scale bud formation? Or that cranial NC migrates that far posterior – and that early?

I did not find a file with legends to the Supplementary material.
Please refer to the literature without using first names (Jan E., Melanie,…)
Although, as a non-native speaker, I think the language is OK, sometimes I feel the wording could be better, and in several cases, an article (‘the’, or ‘a’) is missing.

Reviewer 2 ·

Basic reporting

The study by Germon and colleagues deals with the role of the retinoic acid signaling pathway in tooth and primary placoid scale development in the lesser spotted catshark. This topic is crucial as this pathway has been associated with tooth development in the mouse and fish. More specifically, tinkering with the amount of retinoids has been linked to evolutionary changes in the number of teeth within dentitions and changes in the position of tooth-forming regions potentially reflecting evolutionary transitions of whole dentitions. As teeth likely evolved from dermal denticles, it is plausible to evaluate the role of retinoic acid (RA) pathway in the development of teeth and tooth-like structures (aka “odontodes”) in an animal, where both “odontode” expressions are present. This study conducted on the catshark is thus of high relevance to any further studies dealing with the evolution of odontogenesis.

Application of the all-trans-RA and DEAB pharmacological agents (used in this study) is the standard first step gain- and loss-of-function approach to study the role of RA pathway in the development of any organ/structure, especially in non-models such as the lesser spotted catshark. After the analysis of expression patterns of genes encoding members of the RA pathway, the authors evaluate effects of the pharmacological agents on tooth and scale development to assess differential effect of these agents on tooth/scale development.

The manuscript is very well written, understandable and fluent. My major comment is that the meticulously performed and analyzed pharmacological treatments are relatively poorly introduced and discussed in the context of recent literature. Instead, the discussion conforms to the results section without evaluating a broader aspect of the obtained outcomes (see part 4 General comments).

For minor comments on wordings and typos also see part 4.

Experimental design

The study uses statistical analysis of tooth and scale phenotypes upon abrogation of the RA pathway. From the presented data, I can feel that the authors did their best in this respect. I hail the preliminary treatments that allowed identification of relevant timepoints and concentrations of the applied pharmacological agents prior to conduction of the proper tooth-and-scale-relevant treatments. This is often a neglected part of any experimental setup but may be crucial for the upcoming research using these or other pharmacological agents when studying development of the respective organs/structures. It also allows lowering the number of experimental animals in upcoming experiments (a crucial point in case of the catshark).

As my scientific background does not allow evaluation of the statistics used in the manuscript, I would pass this point on to another reviewer.

I have only a minor question in this respect. The n number in Fig. 2D is not fully clear to me. In Fig. 2D1, n(RA) = 14, but when counting the blue dots in Fig. 2D2, I come to the number 12. Does this mean that the “missing” n number in Fig. 2D2 is due to the absence of any LM/ED/LD tooth germs in the remaining specimens? In other words, does this mean that there were some embryos without any teeth (i.e. completely missing or at EM stage) that are covered in Fig. 2D1 but not in Fig 2D2?

Otherwise, the quality of the chromogenic in situ hybridizations seem to be OK. Although expression patterns of only one retinaldehyde dehydrogenase and four retinoic acid receptors is shown in teeth/scales, the absence of expression of the remaining genes of the RA pathway is clearly justified by their presence in other body parts (Figs. S1 and S3).

Validity of the findings

The findings of the study are important by confirming the role of retinoic acid signaling and adding a substantial knowledge towards understanding of the development of shark teeth and scales. The provided data are robust, understandable and well presented.

In line 325, the authors state that the number of tooth buds was reduced upon application of 10-6 M RA (Fig. 2F1) and that this implies that “RA inhibits tooth initiation during the time of treatment”. However, subsequent histological analysis identified statistically significant change in the number of tooth buds only at LD stage (Fig. 2F2). Wouldn’t it be more plausible to suggest that RA exposure affects only the later stages (ED/LD) of tooth development instead of stating that it inhibits tooth initiation? I guess the authors did not test if any of the earliest markers of tooth development (such as Pitx2, Shh or similar) changed their expression, nor did they analyze any early aspect of tooth development by other means (other than histological, which according to the authors, may be difficult to perform and may be a source of statistical errors)…? If the authors cannot provide such analysis, I am afraid that they cannot claim that exposure to RA leads to failures in tooth initiation (in such case, I would simply change the word “initiation” to “development”). Alternatively, can the authors provide explanation to this concern?

The statement on RA treatment affecting tooth initiation occurs also elsewhere in the text (lines 387, 396, 433) and in the abstract.

In line with this argumentation (lines 398-400), the authors claim that “The missing teeth in treated embryos are teeth at stage LD in control embryos, suggesting that our treatment has an effect only on the initiation stage, and not later (morphogenesis) stages.” Wouldn’t it, again, be more plausible to suggest that the RA treatment affects ED and LD stages (instead of initiation), as these are most affected?

Additional comments

This is an interesting and meticulously performed study performed on precious embryonic material but struggles to introduce the topic and transfer the obtained outcomes of the study to the reader. I suggest these parts being worked on before acceptance of the manuscript for publishing.

In case of the introduction, the authors should work more on and fine-tune its very end. The last sentence stating that “Here we explored this issue by testing the putative function of RA signaling in the development of teeth and scales in a cartilaginous fish, the lesser spotted catshark Scyliorhinus canicula where tooth and scale development has been previously described (tons of citations)” does not help the reader read beyond. It does not qualify or justify why specifically the catshark has been used (although I am sure the authors know why). Also, I am missing a clearly stated research question. Why should the reader care? Why is catshark the most important/relevant/interesting animal for conducting the research on retinoic acid signaling in odontogenesis?

I suggest adding a short paragraph on the catshark itself, i.e. describing development of teeth and caudal primary scales. This will help introduce the model species (either to readers, who have no experience with it, or generally to let the manuscript flow), and guide the reader towards the study itself.

While the suggested changes to introduction are more or less “cosmetical”, the discussion section needs to be worked out more thoroughly. The first (lines 367-385), second (lines 387-402) and fourth (lines 416-424) discussion paragraphs are a mere repetition of the results and virtually do not discuss anything in the context of what is known about RA signaling pathway and tooth development in other vertebrates. Normally, this would not be of a problem, would other parts of the discussion have unfolded them. In the current form, however, they seem only as “shots in the dark”.

How exactly does excess RA in the shark teeth/scales act in comparison to the mouse and zebrafish (lines 432-433)? Is there a difference between expression of RA members in shark versus other vertebrates’ teeth? Are there any differences in expression of factors other than members of the RA pathway between shark tooth buds and caudal primary scales (Berio and Debiais-Thibaud 2020 J Fish Biol)? Could these differences (if there are any relevant) be discussed in the context of what is known from the mouse/fish literature? I am sure the authors can come up with more and more interesting questions that would and should stir the discussion.

I have one more major point related to this section:
Lines 435-437: “Shark teeth are permanently initiated through sequential placode initiation in a dental lamina where stem cells probably develop continuously from cranial neural crest cells (Fraser et al. 2019).”
I am not sure what the authors wanted to state here, but even if Fraser et al. 2019 (a paper comparing properties of human aberrant dental lamina rests and shark dental lamina) did state anything about neural crest cells (which they are not), I am not knowledgeable of any report of cranial neural crest cells contributing to the dental lamina. Can the authors specify what they meant?


Minor points:

Can the authors use “5×10-5” instead of “5.10-5” throughout the text, tables and figures?

Line 64: I suggest removing “therefore” from “The RA signaling pathway has therefore been a labile, quickly evolving…”

Lines 71-75: The authors state that “tooth-like” structures (= odontodes) comprise structures such as dermal denticles of chondrichthyans and pharyngeal denticles of teleosts. To my opinion, teeth are generally considered odontodes as well. However, my point is that the authors do not seem to stick to strict terms. I think that the same structures are named by different terms in the presented manuscript, i.e. the dermal denticles (line 72) are named caudal scales in the manuscript headline and caudal primary scales or dorsal scales throughout the text. I suggest the authors should stick to strict terms defined in the introduction and then use them rigorously throughout the text. Otherwise, the text may sound confusing when using them vaguely while referring to either specific or general meanings of the terms. One such example, line 312, does the term “odontodes” here, means dermal denticles, caudal primary scales only, or teeth + dermal denticles?

Line 90: “medaka and Astyanax mexicanus” is probably better “medaka and the Mexican tetra”

Line 99: suggestion: cite the works of W. E. Reif?

Line 105: “until they reach” change to “until they reached”

Lines 148, 153 and 457: please indicate the correct name of the plasmid used for ligating the gene fragments – does the pSPORTI or Sport1 mean pSV-SPORT1, pCMV-SPORT1 or other?

Line 170: Please consider: “Pharmacological treatment” -> “Pharmacological treatments”

Line 178: “bathes” should be “batches”

Lines 191 and 193: Please consider: “…, and then” or “…, then” -> “…, and the”

Lines 197: Please consider: “dissymmetry” -> “asymmetry”

Line 211: binocular loupe -> binocular scope

Line 243: refence specimens which body length -> refence specimens whose body length

Line 273, 296, 307 and perhaps elsewhere: Can the authors use the word “respectively” at correct places within sentences, i.e. after enumerations of the list of individual cases (and not prior to that)?

Lines 296-297: Please consider editing:
Stage 28 embryos (batches D to F) and late stage 31/early 32 embryos (batches J and K), were exposed continuously to DEAB at respectively 10-4 M (scale exposure) and 10-5 M or 5.10-5M (tooth exposure) during one week (see Table 1 and Table S2).
->
Stage 28 embryos (batches D to F) and late stage 31/early 32 embryos (batches J and K) were exposed to DEAB at 10-4 M (scale exposure) and 10-5 M or 5×10-5 M, respectively, (tooth exposure) continuously for one week (see Table 1 and Table S2).

Lines 299-301: Please consider editing:
In contrast, the number of tooth buds after exposure to 5.10-5 M DEAB was reduced by one-third compared to controls where a mean of 30 tooth buds were observed (Fig 2C1).
->
In contrast, the number of tooth buds was reduced by one third after exposure to 5×10-5 M DEAB as compared to controls, where a mean of 30 tooth buds was observed (Fig 2C1).

Line 311: please consider: was detected at least in scale development, exogenous RA is expected -> was detected during scale development, exogenous RA was expected

Lines 313-314: “RA 10-6 M” -> “10-6 M RA”

Line 322: “although results are not significant” -> “although the results are not statistically significant”

Lines 332-333: “the mean number…was significantly less” -> “the mean number…was significantly lower”

Line 336: general development delay -> general developmental delay

Line 363: “a 5.10-5 M dose” -> “the 5×10-5 M dose”

Line 390: please consider: “had one fourth” -> “possessed only one fourth”

Line 395: Can the authors rephrase the end of the sentence “…but inactivation of tooth buds was much more important.”

Lines 400-402: The authors state that the evaluation of the effect of RA application was not possible because the embryos completely lost scale buds. However, in none of the graphs in Fig 2 do I see a complete loss of scales upon RA treatment. The results show statistically significant change only in the total number of scales upon 10-6 M RA treatment (i.e. by no means a complete loss). Can the authors explain what they meant by the “complete loss”?

Line 416: please consider: pharmaceutical -> pharmacological

Line 417: please consider: by -> due to

Line 438: Can the authors provide citation?

Line 442: in different lineage of vertebrate -> in different lineages of vertebrates

Line 444: please consider: posterior-anterior -> posterior-to-anterior

Line 444: Why do the authors use the term “regeneration”? Does this refer to the regenerating dentitions or to regeneration in general? If the former is the case, it should probably either be well explained or completely omitted as this term (used in relation to tooth renewal or replacement) is not widely accepted among the dental, hard tissue or regeneration communities.


Figures:
Can authors add molarity signs (i.e. change “DEAB 10-4” to “DEAB 10-4 M”) to all headlines of the presented graphs in Fig. 2?

In Fig. 2C1 box plot caption, I think that “DEAB 10-5 M” should be replaced with “DEAB 5×10-5 M”.

Could the authors switch the order of controls and treatments in Fig. S4, so that DMSO controls would be on the left while RA treatments on the right parts of the graphs (i.e. in compliance to other presented graphs)?

---

## Round 0.2 · Minor Revisions

Dear Authors,
I am pleased that both reviewers appreciated your work in revising the manuscript according to their comments. One of the reviewers, in particular, listed an additional series of minor points (for most of them, slight text corrections) that should be easy to introduce. I am therefore looking forward to receive your final version of the edited manuscript.

·

Basic reporting

I am very pleased with the additional explanations on the methods used. I think nevertheless the authors can – and should – emphasize how many jaws were serially sectioned for the tooth counts, as it is an amazing achievement that adds a lot of credibility to their study.
I am also very happy with the extensive comparison, in the discussion, to other species.
Two more small suggestions:
- I think ‘complete lower jaws’ is better English than ‘full lower jaws’ (and perhaps add ‘(i.e. both jaw halves)’ after the first mentioning
- 'First line of teeth' / 'first line teeth': not clear: founder teeth? Or refers to anatomical position?

Experimental design

See under n°1

Validity of the findings

See under n°1

Additional comments

No additional comments

Reviewer 2 ·

Basic reporting

The authors addressed majority of my technical and factual comments and suggestions, and I have no requests other than editing the text, which according to my opinion, still deserves revision. I went through the text and made suggestions for the authors in the most problematic parts of the manuscript. If the authors take time to do these additional edits (although not necessary, I suggest proofreading from a native English speaker), I am happy to let the manuscript be processed further towards the eventual publishing.


Minor points:

Line 31-32: please consider „The general modalities of tooth and "tooth-like" structures (collectively named odontodes) development” -> „The general modalities of development of tooth and "tooth-like" structures (collectively named odontodes)”

Line 39: exposing -> exposure

Line 81: Mice mutated for -> Mouse mutants in

Paragraph in lines 81-111: please stick to either present, or past tense and use it throughout

Lines 94-97: please insert punctuation mark to dissect the two unrelated information present in the sentence

Line 99: lesser spotted catshark -> small-spotted catshark

Lines 102-111: can the authors revise the newly added text, so that it is formally English-correct?
e.g.:
this non-model organism … in this shark species
embryo collection is easy in adult rearing facilities
wide range of sequence data allowing access to gene expression patterns

Line 166: pGEM-T -> pGEM-T Easy

Line 201: the less mortality -> low mortality

Lines 207, 208 and elsewhere: bathes -> baths

Line 214: highest -> higher

Lines 216-217: were transversely serially sectioned as described above for histological staining -> were serially sectioned in a transverse plane for histological staining as described above

Line 221: please consider: before the first tooth buds are developed -> before the appearance of first tooth buds

Line 224: please consider:
Exposure (RA 10-6M or DEAB at 10-5… -> Exposure to pharmacological agents (RA at 10-6M or DEAB at 10-5…) was performed for five days with daily changes of the respective medium. The embryos were then maintained…

Line 234: and a potential source of error -> and may become a potential source of error

Line 240: the full -> whole

Line 263: please consider:
making embryos in the end of the experiment still being at stage 32 -> so that the embryos at the end of the experiments were still at stage 32

Line 270: except gills -> except the gills

Line 273: are -> were

Line 274: extension -> extent

Line 275: have partially regressed -> partially regressed

Line 277: are -> were

Lines 299-314: please stick to either present, or past tense and use it throughout

Lines 343-344: significant batch effect was also observed, most probably … in batch C48 -> significant effect was also observed in batch C48, most probably …

Lines 354: The analysis for scale -> The analysis of scale

Line 355: and only on embryos from batch B -> and on embryos only from batch B

Line 357: at each development stages -> at each developmental stage

Line 366: we tested for -> we analyzed these embryos for

Lines 375: please consider
In RA-treated embryos, embryo body length was distributed between 41 to 56 mm, with half of the embryos less than 49 mm, but developmental scores were mostly greater than 7 (82%), making the treated and control populations not significantly different for developmental score ->
In RA-treated embryos, embryo body length was distributed between 41 to 56 mm, with half of the embryos being shorter than 49 mm. However, developmental scores were mostly greater than 7 (82%), making the treated and control animals not significantly different

Lines 381-383: please consider:
In 10-5M DEAB-treated embryos, more than half were smaller than 49 mm, but the associated total score values were mostly greater than 7 (84 %), and the treated and control populations were not significantly different
->
In 10-5M DEAB treatments, more than half of the embryos was shorter than 49 mm, but the associated total score values were mostly greater than 7 (84 %). Thus, the treated and control populations were not significantly different

Line 385: less than -> shorter than

Line 386: less than -> lower than

Line 386: can the authors rewrite “which resulted”?

Line 388: a DEAB -> the DEAB

Line 389: a DEAB -> the DEAB

Line 401: Our results support that RA-signalling is functional in scale buds -> Our results show that RA-signalling is present in caudal primary scale buds

Lines 402-403: can the authors rewrite the newly added text so that it is formally English correct?

Line 406: the lack of nuclear receptor expression detection in tooth buds -> the lack of RA nuclear receptor expression in tooth buds

Lines 405-407:
However, nuclear receptor levels of expression might be low and under the in situ level of detection but still be active
->
However, expression levels of retinoic acid receptors might be low (under the detection threshold), yet still be active

Line 408: stead expression -> the steady expression

Line 411: teeth or scales -> teeth or caudal primary scales

Line 415: during their morphogenesis -> during morphogenesis

Line 424: “pharyngeal teeth of the mouse”?

Line 435: had -> possessed

Line 439: I do not understand what the authors mean by stating that “the inhibition of tooth bud development was much more important”, please rephrase

Line 450: scale -> caudal primary scale

Line 476: we have evaluated -> we evaluated

Lines 480-481: what do the authors mean by “developmental index”? I was unable to find definition or usage of this term in the methods or elsewhere in the text…

Line 485: can the authors rewrite the following: “This general effect is congruent with the ubiquitous activity of Raldh in the embryo and the important role of RA in its development and in the initiation of a number of functions”
- it is not clear what does “its” refer to
- what did the authors mean by stating “the initiation of a number of functions”?

Lines 487-488: please consider:
as odontoblasts and ameloblasts were shown to be the target of differentiation inhibition by RA signalling
->
as differentiation of odontoblasts and ameloblasts was shown to be affected by low/high (?) levels of RA signaling

Line 492: inhibitory in tooth -> inhibitory for tooth

Line 493: in opposition -> which is in opposition

Lines 495-500: Can the authors rewrite the long sentence, it is unreadable in its current form

Line 505: molar -> molars

Line 507: biological context … are highly diverse -> biological contexts … are highly diverse

Line 513: exogenous RA might be acting as inhibition leverage -> exogenous RA might act as an inhibition leverage

Lines 522-523: it is not clear why the authors propose and make connection between caudal primary scales being patterned by the very early trunk neural crest mesenchyme with their inability to renew

Experimental design

see above

Validity of the findings

see above

Additional comments

see above

---

## Round 0.3 · accepted · Accept

The manuscript has again been very carefully corrected in response to the second round of reviewers' comments and is now clearly suitable for publication.